# Unexpectedly efficient ion desorption of graphene-based materials

Xinming Xia [1,2,3,7], Feng Zhou[4,7], Jing Xu [2,7], Zhongteng Wang[2], Jian Lan[2], Yan Fan[2], Zhikun Wang[2], Wei Liu[2], Junlang Chen [2], Shangshen Feng[2], Yusong Tu [3], Yizhou Yang [5] ✉, Liang Chen [1] ✉ & Haiping Fang [5,6] ✉

Ion desorption is extremely challenging for adsorbents with superior performance, and widely used conventional desorption methods involve high acid or base concentrations and large consumption of reagents. Here, we experimentally demonstrate the rapid and efficient desorption of ions on magnetite-graphene oxide (M-GO) by adding low amounts of $Al^{3+}$. The corresponding concentration of $Al^{3+}$ used is reduced by at least a factor 250 compared to conventional desorption method. The desorption rate reaches ~97.0% for the typical radioactive and bivalent ions $Co^{2+}$, $Mn^{2+}$, and $Sr^{2+}$ within ~1 min. We achieve effective enrichment of radioactive $^{60}Co$ and reduce the volume of concentrated $^{60}Co$ solution by approximately 10 times compared to the initial solution. The M-GO can be recycled and reused easily without compromising its adsorption efficiency and magnetic performance, based on the unique hydration anionic species of $Al^{3+}$ under alkaline conditions. Density functional theory calculations show that the interaction of graphene with $Al^{3+}$ is stronger than with divalent ions, and that the adsorption probability of $Al^{3+}$ is superior than that of $Co^{2+}$, $Mn^{2+}$, and $Sr^{2+}$ ions. This suggests that the proposed method could be used to enrich a wider range of ions in the fields of energy, biology, environmental technology, and materials science.

Adsorption separation technology is one of the most effective and economical separation methods for high-efficiency extraction[1–4], concentration[5–7], and purification[8–11]. Ion-surface adsorption between cations and graphene-based materials[12–18] results in strong adsorption due to the one-atom-layer thickness and perfect aromatic ring structure of graphene[19]. Notably, this strong adsorption of ions can precisely fix the interlayer spacing of graphene membranes for water desalination[20,21], with NaCl crystallization forming on the graphene surface in salt solutions with concentrations far below the saturated concentration[22]. Such strong adsorption can potentially be exploited in graphene-based technology in multiple high-efficiency applications.

For the desorption of ions adsorbed on sorbents, which is one of the most important steps in adsorption separation technology, conventional methods involve the addition of acids and bases, including HCl and NaOH. These methods require a high consumption of HCl or NaOH solutions with concentrations as high as 0.1–0.2 M and a long desorption time of approximately 1–2 h[23–25]. High multivalent metal ions, in particular $Co^{2+}$, $Cu^{2+}$, $Cd^{2+}$, $Cr^{2+}$, and $Pb^{2+}$, strongly interact with

[1]School of Physical Science and Technology, Ningbo University, 315211 Ningbo, China. [2]Department of Optical Engineering, Zhejiang Prov Key Lab Carbon Cycling Forest Ecosy, College of Environmental and Resource Sciences, Zhejiang A&F University, 311300 Hangzhou, China. [3]School of Physical Science and Technology & Microelectronics Industry Research Institute, Yangzhou University, 225009 Yangzhou, China. [4]Radiation Monitoring Technical Center of Ministry of Environmental Protection, State Environmental Protection Key Laboratory of Radiation monitoring, Key Laboratory of Radiation Monitoring of Zhejiang Province, 310012 Hangzhou, China. [5]Department of Physics, East China University of Science and Technology, 200237 Shanghai, China. [6]Wenzhou Institute, University of Chinese Academy of Sciences, 325000 Wenzhou, Zhejiang, China. [7]These authors contributed equally: Xinming Xia, Feng Zhou, Jing Xu. ✉e-mail: yangyizhou@ecust.edu.cn; liangchen@zafu.edu.cn; fanghaiping@ecust.edu.cn

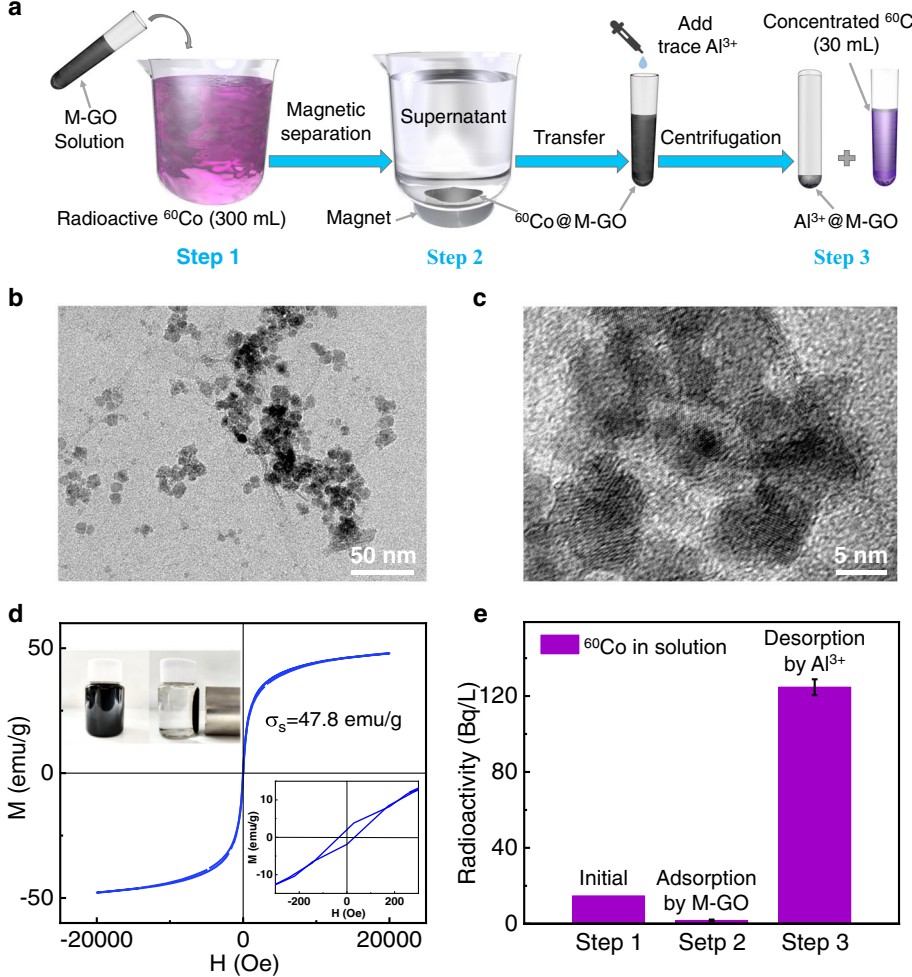

**Fig. 1 | Enrichment of radioactive $^{60}$Co from the solution by the controllable ion adsorption and desorption on M-GO. a** Schematic of $^{60}$Co enrichment. **b** TEM image and **c** high-resolution TEM image of M-GO. **d** Magnetization curve at room temperature (298 K) for the M-GO. Inset is a photograph of separation of M-GO with magnets in solution. **e** Radioactivity of $^{60}$Co at each step of the enrichment experiments. Error bars indicate the standard deviation from three different samples.

the graphene sheet[26] or biosorbents[24,25] and exhibit ineffective or slow desorption[23] using these conventional methods. Moreover, these methods cannot be used to treat some functional graphene-based materials with superior performance, such as magnetite-graphene oxide (M-GO) with $Fe_3O_4$ nanoparticles, because of the simultaneous dissolution of functional groups[27–30]. Therefore, it is difficult to achieve ion desorption with facile, convenient, and low consumption of reagents in graphene-based materials owing to strong ion-surface adsorption. These challenges hinder the potential applications of graphene-based membranes in ion adsorption.

In this study, we observed the unexpectedly rapid and efficient desorption of ions adsorbed on M-GO by adding very low amounts of $Al^{3+}$ (at a volume ratio of 1:500). The desorption rate reaches ~97.0% for typical radioactive and bivalent ions of $Co^{2+}$, $Mn^{2+}$, and $Sr^{2+}$ within ~1 min, yielding a desorption performance superior to that of conventional desorption methods reported to date. Interestingly, we demonstrated the effective enrichment of radioactive $^{60}$Co from the solution by the controllable ion adsorption and desorption on M-GO. The $Al^{3+}$ ions adsorbed on M-GO can be effectively desorbed through the addition of a small amount of $NH_3 \cdot H_2O$. We added 75 µL $NH_3 \cdot H_2O$ (25–28%) to ~30 mL of $Al^{3+}$@M-GO mixture solution (pH ~ 10) and were able to recycle and reuse M-GO without compromising its adsorption efficiency and magnetic performance. Density functional theory calculations revealed that monovalent and divalent ions should have

lower adsorption energies than $Al^{3+}$ adsorption energies[26], suggesting that the proposed method could be used to enrich a wider range of ions.

## Results

M-GO was prepared through chemical co-precipitation of magnetic iron oxide nanoparticles grown on graphene oxide (GO) sheets with $Fe^{3+}$ and $Fe^{2+}$ under alkaline conditions[31,32] (see Supplementary Note 1). $Fe_3O_4$ nanoparticles with an average particle size of 11.6 nm were well distributed in the GO sheets (Fig. 1b). High-resolution transmission electron microscopy (TEM) images (Fig. 1c) and X-ray diffraction (XRD) patterns (Supplementary Fig. 1a) show that the $Fe_3O_4$ nanoparticles have a face-centered cubic structure, with a lattice spacing of 0.257 nm. The $Fe_3O_4$ nanoparticles grown on the GO sheets were further observed using Raman and X-ray photo-electron spectroscopy (XPS) spectra, as shown in Supplementary Fig. 1b and Supplementary Fig. 2, respectively. The magnetization performance of M-GO was measured at room temperature (298 K). As shown in Fig. 1d, the prepared M-GO had high magnetic properties determined by vibrating sample magnetometer (VSM) with a saturation magnetization of 47.8 emu/g. The inset of Fig. 1d shows the easy and rapid M-GO attraction and separation from aqueous solutions, within ~10 min, using an external magnet through magnetic solid/liquid separation.

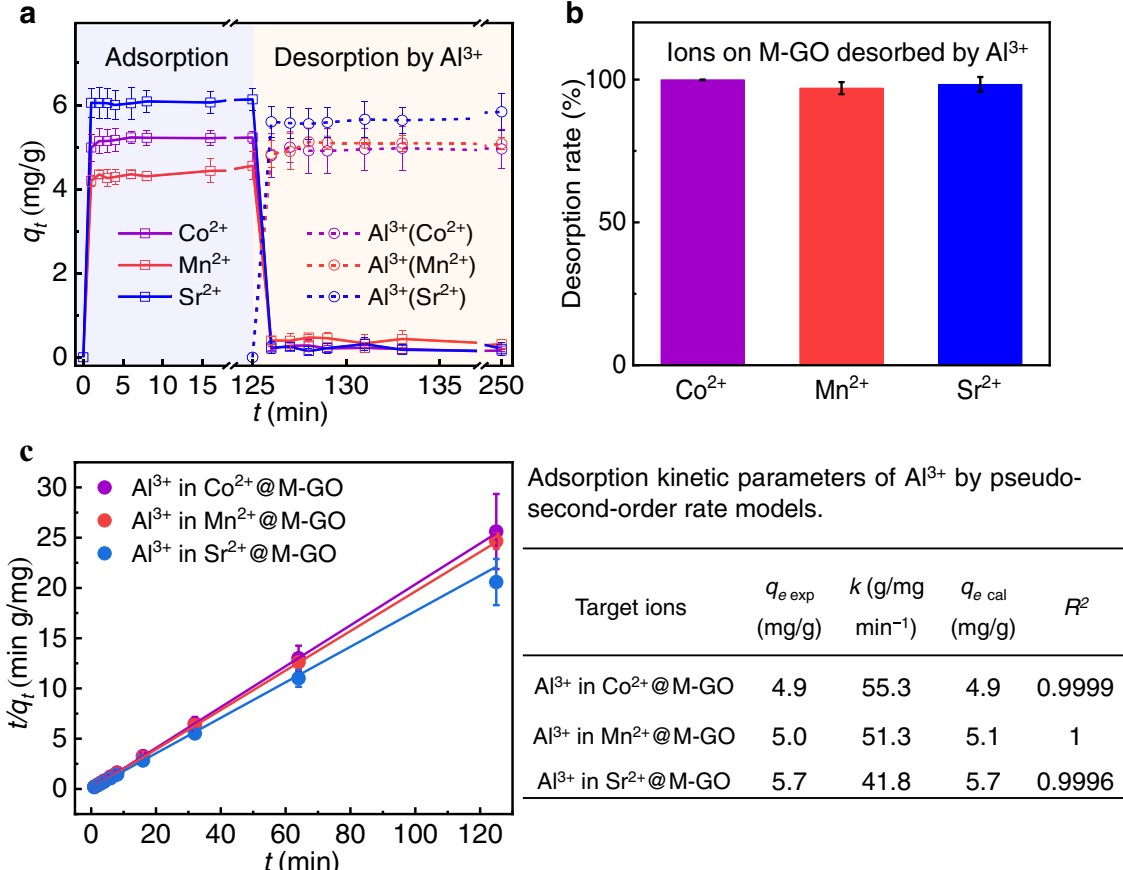

**Fig. 2 | Ion adsorption and desorption of M-GO. a** Adsorption kinetics of 10 mg/L $Co^{2+}$, $Mn^{2+}$, and $Sr^{2+}$ by M-GO, as well as adsorption kinetics of 10 mg/L $Al^{3+}$ added to the salt solutions ($Co^{2+}$, $Mn^{2+}$, and $Sr^{2+}$) at 298 K, respectively. $q_t$ denotes the adsorption capacity of M-GO with time. Light purple and light orange are highlighted to distinguish between adsorption and desorption. **b** Desorption rate of $Co^{2+}$, $Mn^{2+}$, and $Sr^{2+}$ from M-GO by the subsequent addition of $Al^{3+}$. **c** Adsorption kinetic parameters of $Al^{3+}$ during the ion desorption of $Co^{2+}$, $Mn^{2+}$, and $Sr^{2+}$ by a pseudo-second-order rate model. Error bars indicate the standard deviation from three different samples.

Enrichment experiments of radioactive ions using controllable ion desorption on M-GO were performed using radioactive $^{60}Co$ as an example. As illustrated in Fig. 1a, Step 1, the prepared M-GO (60 mg) was added to a 300 mL solution of 15 Bq/L $^{60}Co$ and 1.0 mg/L $Co^{2+}$. In Step 2, the mixtures were stirred at 298 K for 5 min and then separated through magnetic separation. Next, the M-GO, which was adsorbed with $^{60}Co$ and $Co^{2+}$ ions ($^{60}Co@M\text{-}GO$), was removed, and the $^{60}Co@M\text{-}GO$ was dispersed with deionized water to a final volume of 30 mL. In Step 3, 60 μL of $Al^{3+}$ solution was subsequently added such that the concentration of $Al^{3+}$ in the mixtures was 20 mg/L. The solution was stirred at 298 K for 5 min and then separated through magnetic separation and filtration. The radioactivity of $^{60}Co$ in the filtrates was determined using a high-purity germanium γ spectrometer. The results are shown in Fig. 1e; the radioactivity of $^{60}Co$ was only $1.8 \pm 0.4$ Bq/L for the supernatants after magnetic separation in Step 2, showing efficient $^{60}Co$ removal of the M-GO. The final 30 mL solution after desorption exhibited high radioactivity, up to $124.7 \pm 4.1$ Bq/L, and the volume of the solution was reduced by a factor of about 10 compared with the initial $^{60}Co$ solution. This demonstrated the effective enrichment of radioactive elements through controllable ion adsorption and desorption.

We further performed kinetics experiments on the desorption for typical radioactive and bivalent ions of $Co^{2+}$, $Mn^{2+}$, and $Sr^{2+}$ adsorbed by M-GO using $Al^{3+}$ solutions. The prepared M-GO (200 mg) was added to 200 mL solutions of 10 mg/L $Co^{2+}$, $Mn^{2+}$, and $Sr^{2+}$, and the solutions were stirred at 298 K for 125 min. Then, a negligible volume of highly concentrated $Al^{3+}$ solution (400 μL) was subsequently added such that the concentration of $Al^{3+}$ in the mixtures was 10 mg/L. The mixtures

were then stirred at 298 K for another 125 min. At designated time intervals ranging from 0 to 250 min, 5 mL of the solution was taken at each interval for filtration separation and measurement of the residual ion concentration. The adsorption capacities ($q_t$) were calculated (see Supplementary Note 1); the results of the kinetic experiments are shown in Fig. 2a. Rapid ion adsorption of $Co^{2+}$, $Mn^{2+}$, and $Sr^{2+}$ adsorbed by M-GO occurred within 1 min after the addition of ions. The adsorption capacities ($q_t$) and the equilibrium adsorption capacities ($q_e$) of M-GO (Supplementary Fig. 3) for $Co^{2+}$, $Mn^{2+}$, $Sr^{2+}$, and $Al^{3+}$ solutions are consistent with those of the previous reports[31–33]. Interestingly, for the subsequent addition of 10 mg/L $Al^{3+}$ ions at 125 min, there was a thorough desorption of the $Co^{2+}$, $Mn^{2+}$, and $Sr^{2+}$ ions that were originally adsorbed on M-GO, along with the corresponding rapid adsorption of $Al^{3+}$ ions. The desorption rate by the addition of $Al^{3+}$ ions reached $99.9 \pm 0.1\%$, $97.0 \pm 2.1\%$, and $98.3 \pm 2.6\%$ for $Co^{2+}$, $Mn^{2+}$, and $Sr^{2+}$ solutions, respectively (Fig. 2b).

We analyzed the kinetic parameters of the ion desorption. Considering that the ion desorption occurred via $Al^{3+}$ ion substitution, the desorption kinetic parameters of $Co^{2+}$, $Mn^{2+}$, and $Sr^{2+}$ can be estimated by the adsorption kinetic parameters of $Al^{3+}$ during the desorption processes. A pseudo-second-order rate equation[34], which has been widely applied to the adsorption of graphene-based materials[27], was applied in the ion desorption via $Al^{3+}$ ion substitution as follows:

$$\frac{t}{q_t} = \frac{1}{kq_e^2} + \frac{1}{q_e}t \tag{1}$$

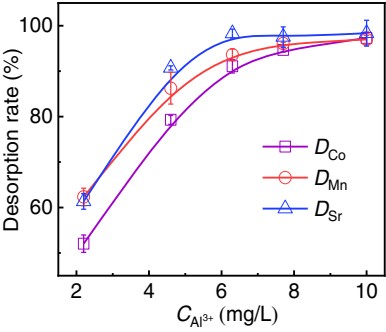

**Fig. 3 | Desorption rate for Co²⁺, Mn²⁺, and Sr²⁺ ions on M-GO analyzed by addition of Al³⁺.** Error bars indicate the standard deviation from three different samples.

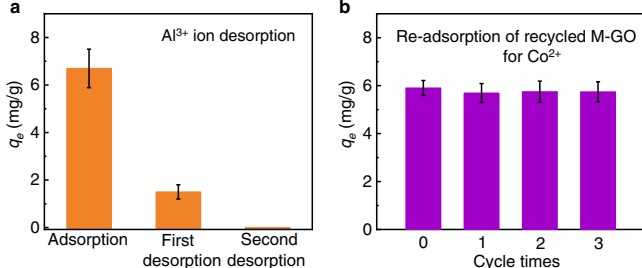

**Fig. 4 | Al³⁺ ion desorption and reusability of M-GO for Co²⁺ adsorption.** **a** Desorption of Al³⁺ on Al³⁺@M-GO solution. **b** Re-adsorption of recycled M-GO for 10 mg/L of Co²⁺. Error bars indicate the standard deviation from three different samples.

where $k$ (g/mg min⁻¹) is the equilibrium rate constant of the pseudo-second-order rate for Al³⁺, $q_t$ (mg/g) is the amount of Al³⁺ adsorbed on the M-GO at time $t$ (min), and $q_e$ (mg/g) is the adsorption capacity at equilibrium.

As shown in Fig. 2c, the calculated adsorption capacities ($q_{e\,cal}$) are consistent with the corresponding experimental values ($q_{e\,exp}$), and the $R^2$ for the linear plots are close to 1, indicating that the kinetic adsorption can be well described by the pseudo-second-order rate equation. Remarkably, the $k$ values of Al³⁺ during the desorption of the Co²⁺, Mn²⁺, and Sr²⁺ ions are 55.3, 51.3, and 41.8 g/mg min⁻¹, respectively, which are about two or three orders of magnitude higher than the equilibrium rate constants of other types of adsorbents, including zeolites, zinc ferrite nanoparticles, biochar, GO-based membranes, and polymeric adsorbents (Supplementary Table 1). The results indicate the ultrafast Al³⁺ ion substitution adsorption, as well as the simultaneous ion desorption of Co²⁺/Mn²⁺/Sr²⁺ on M-GO. Furthermore, similar rapid desorption of the Cu²⁺ and Cd²⁺ ions also can be achieved by our method (see Supplementary Note 4), showing a wide range of applications of the method in this work.

In addition, we analyzed the adsorption kinetics of mixed Co²⁺, Mn²⁺, and Sr²⁺ salt solutions by M-GO and the desorption kinetics of Al³⁺. A similar rapid ion adsorption of the mixed solution adsorbed by M-GO occurred within 1 min (Supplementary Fig. 5). The corresponding equilibrium adsorption capacities were 3.4 ± 0.1, 1.3 ± 0.1, and 2.0 ± 0.1 mg/g for Co²⁺, Mn²⁺, and Sr²⁺, respectively. With the subsequent addition of 10 mg/L Al³⁺ ions at 60 min, the thorough desorption of Co²⁺, Mn²⁺, and Sr²⁺ ions originally adsorbed on M-GO occurred within ~1 min, along with the corresponding rapid adsorption of Al³⁺ ions. The total desorption rate for all mixed ions was 98.6 ± 1.3%, which corresponds to desorption rates of 98.6 ± 1.6%, 99.9 ± 0.1%, and 97.3 ± 4.7% for Co²⁺, Mn²⁺, and Sr²⁺, respectively. Therefore, the rapid adsorption, especially the efficient mixed ions desorption on M-GO by adding very low amounts of Al³⁺, is still consistent with those of single-salt solutions.

## Discussion

Recent studies reported that ion adsorption equilibrium was achieved in 20–30 min for GO membranes[35–37]. In contrast, the efficient adsorption equilibrium of M-GO was achieved within 1 min, which was attributed to the large specific surface area and high dispersibility of the M-GO sheets in the solution. Notably, Co²⁺, Mn²⁺, and Sr²⁺ ions that are efficiently adsorbed on M-GO can be effectively desorbed by the addition of Al³⁺ ions at a concentration of less than 0.4 mM (10 mg/L Al³⁺ in mixture solutions). It is important to note that graphene-based materials[22,38] and biosorbents[24,25] exhibit strong ion adsorption, and high multivalent metal ions, such as Co²⁺, Cu²⁺, Cd²⁺, Cr²⁺, and Pb²⁺, interact particularly strongly with graphene sheets[16] or biosorbents[24,25]. Conventional methods for the desorption of these ions require high

volumes of highly concentrated (0.1–0.2 M) acids and bases, such as HCl and NaOH[24,25], and cannot be used to treat M-GO because of the simultaneous dissolution of the Fe₃O₄ nanoparticles present. Thus, our results demonstrate the rapid and thorough desorption of Co²⁺, Mn²⁺, and Sr²⁺ ions on M-GO through the addition of Al³⁺ ions. Remarkably, the eluted concentration of Al³⁺ was reduced by a factor of at least 250 compared with the conventional desorption method.

In addition, we analyzed the effects of Al³⁺ concentration on the ion desorption of Co²⁺, Mn²⁺, and Sr²⁺ on M-GO. As shown in Fig. 3, there was significant desorption of 40–60% for Co²⁺, Mn²⁺, and Sr²⁺ ions (10 mg/L in mixtures), even with the addition of a very limited amount of Al³⁺ (~2 mg/L in mixtures). The desorption rate increased with increasing concentration of Al³⁺ ions and reached ~95% when ~8 mg/L of Al³⁺ was added. The results further confirmed the efficient ion desorption on M-GO by our method of Al³⁺ ion treatment.

From the cycle sustainability of Al³⁺@M-GO, the higher trivalent metal ions of Al³⁺ interact strongly with the M-GO sheet over other bivalent metal ions, which introduced a more difficult desorption. Fortunately, we found that the Al³⁺ ions adsorbed on M-GO can be effectively desorbed by adding a small amount of NH₃·H₂O. In detail, 75 µL NH₃·H₂O (25-28%) was added to the 30 mL mixture solutions to adjust the pH to 10, and then the mixtures were separated through magnetic separation and filtration. The separated M-GO was desorbed again with 30 mL DI water (containing 75 µL NH₃·H₂O). The concentrations of Al³⁺ in the filtrates were determined. The desorption rates of Al³⁺ on M-GO reached 78.5 ± 4.0% and 99.9 ± 0.1% for the two desorption steps, indicating the achievement of recycled M-GO. However, when adding the same amount of NH₃·H₂O to Co²⁺@M-GO, the desorption of Co²⁺ ions cannot be achieved. Importantly, the recycled M-GO can be reused easily multiple times without compromising its adsorption efficiency and magnetic performance (Fig. 4 and Supplementary Fig. 6). Furthermore, the corresponding concentration of NH₃·H₂O used here was two to three orders of magnitude smaller than the conventional acid-base desorption method.

We noted that alkaline conditions generally increase the ion adsorption of GO[31,32]. However, Al³⁺ is unique under alkaline conditions, where aluminate anion [Al(OH)₄]⁻ will be the dominant species at pH 10[39,40]. Such anionic species would exhibit weak adsorption or repulsion to an electronegative π-conjugated system, including graphene, GO, and other materials composed of aromatic rings[15,22,26].

We further performed quantum chemistry calculations to elucidate the underlying physical mechanism occurring on the surface of graphene. We used the hydrocarbon C₆₈H₂₂ as a model for graphene and calculated the corresponding hydrated cation adsorbed complexes X@G (G = C₆₈H₂₂, X = Al³⁺-(H₂O)₆, Co²⁺-(H₂O)₆, Mn²⁺-(H₂O)₆, and Sr²⁺-(H₂O)₆) at the level of M06-2X/Def2-SVP. Here, Al³⁺-(H₂O)₆@G and Co²⁺-(H₂O)₆@G are chosen as examples, and Fig. 5 shows their structures, cation adsorption distances, cation partial charges, electron density differences, and adsorption energies. For Mn²⁺-(H₂O)₆@G and

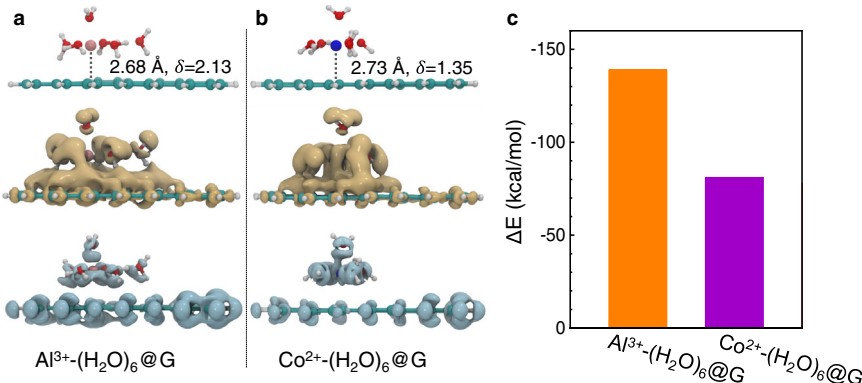

**Fig. 5 | Theoretical computations for ions on graphene.** The most stable optimized geometries and electron density differences of X@G complexes: **a** X is $Al^{3+}$-$(H_2O)_6$, **b** X is $Co^{2+}$-$(H_2O)_6$. Spheres in green, white, and red represent carbon, hydrogen, and oxygen atoms, respectively. Pink and blue spheres represent $Al^{3+}$ and $Co^{2+}$, respectively. Adsorption distances (in Å) and partial charges of cation $\delta$ (in atomic units) are listed. The increased and decreased electron densities (from −0.1 to 0.1) are in khaki and gray, respectively. **c** The calculated adsorption energies of X@G at the level of M06-2X/Def2-SVP.

$Sr^{2+}$-$(H_2O)_6$@G, the corresponding results are shown in Supplementary Fig. 7. Calculation results show that all hydrated ions can be stably adsorbed on the surface of G, and the adsorption distances range from 2.38 Å to 2.73 Å. The adsorption energies of $Co^{2+}$-$(H_2O)_6$@G, $Mn^{2+}$-$(H_2O)_6$@G, and $Sr^{2+}$-$(H_2O)_6$@G are very close (around −80 kcal/mol), while the adsorption energy of $Al^{3+}$-$(H_2O)_6$@G is approximately 75% higher (−139 kcal/mol). This is also supported by the results for the cation partial charges ($\delta$) and electron density differences of X@G, showing that $Al^{3+}$ leads to the greatest reduction in partial charges and the greatest increase in electron densities mainly transferred from $C_{68}H_{22}$. Clearly, these results reveal the strong advantages of $Al^{3+}$ ion adsorption on M-GO compared to $Co^{2+}$, $Mn^{2+}$, and $Sr^{2+}$ ions. Here, the adsorption energy is mainly due to the interaction between the hydrated cation and the aromatic rings in graphene, namely the hydrated cation-π interaction[20,26,41]. The existence of these interactions was confirmed by ultraviolet absorption spectroscopy (Supplementary Fig. 8).

We can estimate the adsorption probability of $Al^{3+}$ on graphene ($P_{Al}$), relative to that of $Co^{2+}$ ($P_{Co}$), as follows:

$$\frac{P_{Al}}{P_{Co}} = \exp\frac{\Delta E_{Co} - \Delta E_{Al}}{k_B T} \qquad (2)$$

where $k_B$ is Boltzmann's constant and $T$ is the temperature. At 300 K, the calculated $P_{Al}/P_{Co}$ is $1.03 \times 10^{42}$. This result demonstrates that the adsorption probability of $Co^{2+}$ is completely negligible compared to that of $Al^{3+}$, indicating that the $Co^{2+}$ ions adsorbed on graphene can be rapidly desorbed by $Al^{3+}$ ions; this is consistent with our experimental observations. Considering that universal monovalent and divalent ions should have smaller adsorption energies than those of $Al^{3+}$ [26], we suggest that the method proposed in the present study could be used to enrich a wider range of ions.

In summary, we have experimentally achieved effective ion ($Co^{2+}$, $Mn^{2+}$, and $Sr^{2+}$) adsorption and desorption of M-GO by adding very low amounts of $Al^{3+}$. Unlike conventional desorption methods that use large amounts of HCl or NaOH solutions with high concentration, our desorption method involving the addition of very low amounts of $Al^{3+}$ is facile and convenient and consumes low amounts of reagent. Importantly, we demonstrated the effective enrichment of radioactive $^{60}Co$ from the solution by the controllable ion adsorption and desorption on M-GO. Density functional theory calculations indicate that these facile adsorption and desorption processes originate from the hydrated cation-π interaction between the ions and the π-conjugated system in the graphitic surface, which promotes ion-surface adsorption and accounts for the huge difference in adsorption probability

between $Al^{3+}$ ions and other ions. Notably, based on the unique hydrolysis of $Al^{3+}$, the M-GO can be conveniently recycled and easily reused multiple times without compromising its adsorption efficiency and magnetic performance. We also noted that monovalent and divalent ions should have lower adsorption energies than that of $Al^{3+}$, indicating that this method could be used for the adsorption and desorption of a wider range of ions. Thus, these findings represent a facile step for the high-efficiency desorption, extraction, and concentration of ions with potential applications, including nuclear energy, medicine, agriculture, and nuclear wastewater treatment.

## Methods
### Experimental operations
GO was prepared from natural graphite powder using a modified Hummers method[42]. M-GO was prepared through chemical co-precipitation of magnetic iron oxide nanoparticles by coating GO with $Fe^{3+}$ and $Fe^{2+}$ under alkaline conditions[31,32] and was characterized by TEM, XRD, Raman spectroscopy, XPS, and VSM (see Supplementary Note 1). In the adsorption and desorption of radioactive $^{60}Co$ for enrichment, the radioactive $^{60}Co$ solution was prepared from a $^{60}Co$ standard solution (National Institute of Metrology). The activity concentration of $^{60}Co$ in the solution was determined using a high-purity germanium γ spectrometer (GEM-100). The enrichment experiment was performed according to the steps shown in Fig. 1a. The activity concentrations of $^{60}Co$ in the adsorbed and desorbed solutions (Steps 2 and 3, respectively) were detected after magnetic separation and filtration, respectively. Supplementary Note 1 and 4 provide details on the ion adsorption and desorption of M-GO and the reusability of M-GO for $Co^{2+}$ adsorption.

### Theoretical calculations
The M06-2X[43] method and Def2-SVP basis set[44] were employed for geometric optimization, frequency, and energy calculations. Both the low-spin state and high-spin state for $Co^{2+}$ and $Mn^{2+}$ systems were considered. All minima have no virtual frequency. The adsorption energies ($\Delta E_X$) are defined as follows:

$$\Delta E = E_{X@G} - E_G - E_X \qquad (3)$$

where $E_{X@G}$ denotes the total energy of the cation/hydrated cation adsorbed on graphene, and $E_G$ and $E_X$ denote the energies of the isolated graphene and the cation/hydrated cation, respectively. Partial charges ($\delta$) at the M06-2X/def2-SVP level were calculated using natural bond orbital (NBO) analysis[45,46]. All electronic calculations were performed using the Gaussian 16 program package[47]. The electron density

differences were analyzed using the Multiwfn program[48], and the structures were visualized using VMD[49].

## Data availability

The authors declare that all the data supporting the findings of this study are available within the article (and its Supplementary Information file), or available from the corresponding author.

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

## Acknowledgements

This work was supported by the National Natural Science Foundation of China (12074341, U1832150, 11875236, 11975206, 12075211, 11905186), the Fundamental Research Funds for the Provincial Universities of Zhejiang (2020TD001), the Scientific Research and Developed Fund of Zhejiang A&F University (No. 2017FR032), and the Scientific Research and Developed Funds of Ningbo University (No. ZX2022000015).

## Author contributions

H.F., L.C., and Y.Y. conceived the ideas. L.C., X.X., F.Z., S.F., and Y.Y. designed the experiments, simulations and co-wrote the manuscript. X.X., F.Z., Y.F., Z.W., and J.L. performed the experiments and prepared the data graphs. F.Z. performed the radioactive $^{60}$Co enrichment experiments. J.X, Z.W., Y.Y., W.L., J.C., and Y.T. performed the simulations. All authors discussed the results and commented on the manuscript.

## Competing interests

The authors declare no competing interests.
