## [Peer Review File · Nature Communications]

Reviewers' comments:

Reviewer #1 (Remarks to the Author):

In this manuscript, the authors studied the desorption of Co(II), Mn(II) and Sr(II) from M-GO by the addition of Al³⁺. The kinetic desorption and DFT calculation about the interaction of Al³⁺ and other metal ions with M-GO were also carried out to understand the desorption of Co(II), Mn(II) and Sr(II). The kinetic desorption of bivalent metal ions from GO or M-GO by adding trivalent metal ions is a general information. The DFT calculation only gave the adsorption energies simply. Based on the contents, I think the manuscript does not merit the publication requirement in Nature Communication.

Special comments:

1. There are some grammar errors in the main text such as "Cr²⁺" etc.
2. The authors mentioned "unexpectedly rapid and efficient desorption", it is necessary to give the desorption kinetic parameters to understand the rapid desorption.
3. The binding of Al³⁺, Co(II), Mn(II) and Sr(II) on M-GO should be calculated in detail, especial the bond distance, the electronic density etc. The theoretical calculation about the kinetic sorption of Al³⁺ on M-GO should be added to understand the rapid sorption of Al³⁺ on M-GO.
4. It is very strange that the sorption of Sr²⁺ is the highest, but the desorption percentage is also the highest.

Reviewer #2 (Remarks to the Author):

This paper by Chen and co-workers reports efficient desorption of adsorbed metal ions on magnetite-graphene oxide (M-GO) by addition of Al³⁺ ions. Usefulness of the reported method has been demonstrated by concentration of radioactive ⁶⁰Co through simple filtration and addition of Al³⁺ ions processes. These findings are of interest, but the biggest concern from a viewpoint of practical use and sustainability is the reusability of M-GO materials after desorption with Al³⁺ ions. As indicated by competition experiments and theoretical calculations, Al³⁺ ions strongly bind onto M-GO over other metal ions such as Co²⁺. Then, how can adsorbed Al³⁺ ions be desorbed from M-GO? If it requires addition of a large amount of HCl or NaOH, a desorption problem that the authors claimed in the introduction section will occur again. In addition, desorption selectivity upon addition

of Al³⁺ ions should be discussed. Namely, analyze desorption amount of metal ions using M-GO which adsorbed two (or more) different metal ions.

Other points:

(1) Phrases such as “adding trace amounts of Al³⁺” (page 3, line 4), “a negligible volume of Al³⁺” (page 5, line 7) are overstatements. Analytically, these amounts are not negligible.

(2) References cited in the introduction section are a bit biased. The authors cite only uranium examples for general high-efficiency extraction and concentration. Furthermore, references 6–10 are papers discussing cation- π interactions or noncovalent interactions, but not showing graphene-based materials.

In the present state, this manuscript is not suitable for publication in Nature Communications. I recommend a more specialized journal to be re-submitted after addressing comments above.

Reviewer #3 (Remarks to the Author):

I recommend rejecting this paper based on poor English grammar usage. The manuscript in its current form is substandard. I am not commenting on the scientific merits of the research; rather, I am unable to understand the paper sufficiently to judge its scientific merits. The authors are encouraged to submit this manuscript for professional English technical editing by a fluent speaker.

Responses to Referees' Comments

Manuscript ID: NCOMMS-21-21067

Title: "Unexpectedly efficient ion desorption of graphene-based materials"

Referee #1 (Remarks to the Author):

In this manuscript, the authors studied the desorption of Co (II), Mn (II) and Sr (II) from M-GO by the addition of Al³⁺. The kinetic desorption and DFT calculation about the interaction of Al³⁺ and other metal ions with M-GO were also carried out to understand the desorption of Co (II), Mn (II) and Sr (II). The kinetic desorption of bivalent metal ions from GO or M-GO by adding trivalent metal ions is a general information. The DFT calculation only gave the adsorption energies simply. Based on the contents, I think the manuscript does not merit the publication requirement in Nature Communication.

Reply: We sincerely thank the referee for his/her very helpful suggestions and very constructive comments.

As the referee said, it is general information that the interaction of graphene with Al³⁺ is much stronger than that with divalent ions. However, the highlights of our work are mainly reflected in two aspects. First, one of the chief advantages is the efficient desorption of ions on M-GO; that is, the desorption rate of typical bivalent ions of Co²⁺, Mn²⁺, and Sr²⁺ can reach 97.0% by adding very low amounts of Al³⁺ (with a volume ratio of 1:500, 10 mg/L). The desorption performance is superior to that of the conventional desorption methods reported to date, which usually require the addition of acids and bases, including HCl and NaOH, and consume large amounts of HCl or NaOH solutions with concentrations as high as 0.1~0.2 M. The second chief advantage is the ultrafast ion desorption, within ~1 min, which is superior to that of conventional desorption methods with a long desorption time of approximately 1~2 h [*Water Res.* **39**, 2273-2280 (2005); *Chem. Eng. J.* **151**, 113-121 (2009)]. Such rapid and efficient desorption or adsorption of ions on M-GO is unexpected and attributed to their superior dispersibility and the high specific surface area of monolayer GO in water [*Carbon* **52**, 171-180, (2013); *Chem. Eng. J.* **226**, 336-347 (2013)] as well as the strong cation- π interactions of the cations with the aromatic rings in the GO surfaces [*Nature* **550**, 380-383 (2017)].

In response to the referee's suggestions, the revised manuscript reports on new DFT calculations of the binding of the cations on M-GO, including optimized geometries, energy adsorption distances, partial charges, and electron density differences. The new DFT calculation results show that all hydrated ions can be stably adsorbed on the surface of the M-GO model with considerable adsorption energies, and Al³⁺ has the strongest interaction with graphene compared with other cations studied here. In addition, *ab initio* molecular dynamic (AIMD) simulations were also

performed to illustrate the underlying physical mechanism taking place in the kinetic sorption of Al^{3+} on M-GO. The AIMD results of adsorption time and mean square displacements (MSD) show a fast adsorption speed of Al^{3+} to graphene.

Further and importantly, we have achieved the reuse of M-GO by Al^{3+} ion desorption in our revised work. From the cycle sustainability of Al^{3+} @M-GO, the higher trivalent metal ions of Al^{3+} interact strongly with the M-GO sheet over other bivalent metal ions, which introduced a more difficult desorption. Surprisingly, we found that the Al^{3+} ions adsorbed on M-GO can be effectively desorbed by the addition of a small amount of $\text{NH}_3\cdot\text{H}_2\text{O}$, such as 75 μL $\text{NH}_3\cdot\text{H}_2\text{O}$ (25~28%) added to the ~30 mL of Al^{3+} @M-GO mixture solution (pH ~10). Yet, when the same amount of $\text{NH}_3\cdot\text{H}_2\text{O}$ was added to Co^{2+} @M-GO, the desorption of Co^{2+} ions cannot be achieved. Importantly, the recycled M-GO could be reused easily multiple times without compromising its adsorption efficiency and magnetic performance, using Co^{2+} as an example as shown below in Fig. R1.

We noted that the alkaline conditions generally increase the ion adsorption of GO [J. Phys. Chem. C **115**, 25234-25240 (2011); ACS Appl. Mater. Interfaces **6**, 9871-9880 (2014)]. However, Al^{3+} is unique under alkaline conditions, where aluminate anion $[\text{Al}(\text{OH})_4]^-$ is the dominant species at pH 10 [Sci. Total Environ. **662**, 1003–1011 (2019); Int. J. Environ. Anal. Chem. **86**, 1007–1018 (2006)]. Such anionic species would exhibit weak adsorption or repulsion to an electronegative π -conjugated system, including graphene, GO, and other materials composed of aromatic rings [Chem. Rev. **97**, 1303-1324 (1997); Nat. Chem. **10** (7), 776-779 (2018); Sci. Rep. **3**, 3436 (2013)].

Fig. R1. Al^{3+} ion desorption and reusability of M-GO for Co^{2+} adsorption. (a) Desorption of Al^{3+} on Al^{3+} @M-GO solution. (b) Re-adsorption of recycled M-GO for 10 mg/L of Co^{2+} . Error bars indicate the standard deviation from three different samples.

Special comments:

1. There are some grammar errors in the main text such as “Cr²⁺” etc.

Reply: We thank the referee for his/her comments.

Our revised manuscript has been edited by a professional English editing service.

There are two types of chromium ions, Cr²⁺ and Cr³⁺. We checked the reference we cited in the main text [*Sci. Rep.* **3**, 3436 (2013)], in which adsorption energy between Cr²⁺ and graphene was studied.

2. The authors mentioned “unexpectedly rapid and efficient desorption”, it is necessary to give the desorption kinetic parameters to understand the rapid desorption.

Reply: We thank the referee for his/her constructive suggestions.

According to the referee's suggestions, we further analysed the kinetic parameters of the ion desorption. Considering the ion desorption via Al³⁺ ion substitution, the desorption kinetic parameters of Co²⁺, Mn²⁺, and Sr²⁺ can be estimated by the adsorption kinetic parameters of Al³⁺ during the desorption processes. A pseudo-second-order rate equation [*J. Am. Chem. Soc.* **139**, 12745–12757 (2017)], which has been widely applied to the adsorption of graphene-based materials [*ACS Nano* **4**, 3979–3986 (2010)], was applied in the ion desorption via Al³⁺ ion substitution, as follows:

$$\frac{t}{q_t} = \frac{1}{kq_e^2} + \frac{1}{q_e}t \quad (1)$$

where k (g/mg min⁻¹) is equilibrium rate constant of the pseudo-second-order rate for Al³⁺, q_t (mg/g) is the amount of Al³⁺ adsorbed on the M-GO at time t (min), and q_e (mg/g) is the adsorption capacity at equilibrium.

Fig. R2. Adsorption kinetic parameters of Al³⁺ during the ion desorption of Co²⁺, Mn²⁺, and Sr²⁺ by a pseudo-second-order rate model.

As shown in Fig. R2, the calculated adsorption capacities ($q_{e \text{ cal}}$) are consistent

with the corresponding experimental values ($q_{e \text{ exp}}$), and the R^2 for the linear plots are close to 1, indicating that kinetic adsorption can be well described by the pseudo-second-order rate equation. Remarkably, the k values of Al^{3+} during the ion desorption of Co^{2+} , Mn^{2+} , and Sr^{2+} are 55.3, 51.3, and 41.8 g/mg min^{-1} , respectively, which are about two or three orders of magnitude higher than the equilibrium rate constants of other types of adsorbents, including zeolites, zinc ferrite nanoparticles, biochar, GO-based membranes, and polymeric adsorbents (Table R1). The results indicate the ultrafast Al^{3+} ion substitution adsorption, as well as the simultaneous ion desorption of Co^{2+} , Mn^{2+} , and Sr^{2+} on M-GO.

Table R1. Comparison of the equilibrium rate constant (k) of the pseudo-second-order rate on different adsorbents in solutions for multivalent metal ions in the literature.

Adsorbents	Target ions	k (g/mg min^{-1})	References
ZIF-8	Cu^{2+}	2.81×10^{-1}	f1
ZF-NPS	Cd^{2+}	1.65×10^{-1}	f2
BC@MnO ₂	Cu^{2+}	4.60×10^{-3}	f3
	Cu^{2+}	5.58×10^{-3}	f4
MoS ₄ -LDH	Pb^{2+}	2.71×10^{-1}	f4
	Hg^{2+}	3.62×10^{-1}	f4
Chitosan	Al^{3+}	1.68×10^{-3}	f5
GO membranes	Cu^{2+}	1.55×10^{-2}	f6
	Cd^{2+}	1.79×10^{-2}	f6
	Ni^{2+}	1.05×10^{-2}	f6
GO/cellulose membranes	Co^{2+}	9.40×10^{-3}	f7
	Zn^{2+}	9.70×10^{-3}	f7
	Pb^{2+}	5.50×10^{-3}	f7
VA-PG	UO_2^{2+}	9.10×10^{-7}	f8
GO-MnFe ₂ O ₄	Pb^{2+}	9.50×10^{-1}	f9
PANI/GO	Sr^{2+}	9.50×10^{-3}	f10
MGO-PP	UO_2^{2+}	1.90×10^{-3}	f11
M-RGO	As^{3+}	5.00×10^{-2}	f12
	Co^{2+}	1.86×10^{-2}	f13
	Cu^{2+}	1.10×10^{-3}	f14
	Fe^{2+}	47.2	f15
	Mn^{2+}	43.6	f15
M-GO	Al^{3+} (in Co^{2+} solution)	55.3	this work
	Al^{3+} (in Mn^{2+} solution)	51.3	this work
	Al^{3+} (in Sr^{2+} solution)	41.8	this work

- f1. Latrach, Z., Moumen, E., Kounbach, S. & El Hankari, S. Mixed-Ligand Strategy for the Creation of Hierarchical Porous ZIF-8 for Enhanced Adsorption of Copper Ions. *ACS Omega* **7**, 15862–15869 (2022).
- f2. Zhao, X., Baharinikoo, L., Farahani, M. D., Mahdizadeh, B. & Farizhandi, A. A. K. Experimental modelling studies on the removal of dyes and heavy metal ions using ZnFe₂O₄ nanoparticles. *Sci. Rep.* **12**, 5987 (2022).
- f3. Zhang, H. et al. Enhanced removal of heavy metal ions from aqueous solution using manganese dioxide-loaded biochar: Behavior and mechanism. *Sci. Rep.* **10**, 1–13 (2020).
- f4. Ma, L. et al. Highly Selective and Efficient Removal of Heavy Metals by Layered Double Hydroxide Intercalated with the MoS₄²⁻ Ion. *J. Am. Chem. Soc.* **138**, 2858–2866 (2016).
- f5. Septhum, C., Rattanaphani, S., Bremner, J. B. & Rattanaphani, V. An adsorption study of Al(III) ions onto chitosan. *J. Hazard. Mater.* **148**, 185–191 (2007).
- f6. Tan, P. et al. Adsorption of Cu²⁺, Cd²⁺ and Ni²⁺ from aqueous single metal solutions on graphene oxide membranes. *J. Hazard. Mater.* **297**, 251–260 (2015).
- f7. Sitko, R., Musielak, M., Zawisza, B., Talik, E. & Gagor, A. Graphene oxide/cellulose membranes in adsorption of divalent metal ions. *RSC Adv.* **6**, 96595–96605 (2016).
- f8. Liu, T. et al. Vertically Aligned Polyamidoxime/Graphene Oxide Hybrid Sheets' Membrane for Ultrafast and Selective Extraction of Uranium from Seawater. *Adv. Funct. Mater.* **32**, 2111049 (2022).
- f9. Kumar, S. et al. Graphene Oxide–MnFe₂O₄ Magnetic Nanohybrids for Efficient Removal of Lead and Arsenic from Water. *ACS Appl. Mater. Interfaces* **6**, 17426–17436 (2014).
- f10. Hu, B. et al. Decontamination of Sr(II) on Magnetic Polyaniline/Graphene Oxide Composites: Evidence from Experimental, Spectroscopic, and Modeling Investigation. *ACS Sustainable Chem. Eng.* **5**, 6924–6931 (2017).
- f11. Dai, Z., Sun, Y., Zhang, H., Ding, D. & Li, L. Highly Efficient Removal of Uranium(VI) from Wastewater by Polyamidoxime/Polyethyleneimine Magnetic Graphene Oxide. *J. Chem. Eng. Data* **64**, 5797–5805 (2019).
- f12. Chandra, V. et al. Water-Dispersible Magnetite-Reduced Graphene Oxide Composites for Arsenic Removal. *ACS Nano* **4**, 3979–3986 (2010).
- f13. Liu, M., Chen, C., Hu, J., Wu, X. & Wang, X. Synthesis of Magnetite/Graphene Oxide Composite and Application for Cobalt(II) Removal. *J. Phys. Chem. C* **115**, 25234–25240 (2011).
- f14. Li, J. et al. Removal of Cu(II) and Fulvic Acid by Graphene Oxide Nanosheets Decorated with Fe₃O₄ Nanoparticles. *ACS Appl. Mater. Interfaces* **4**, 4991–5000 (2012).

f15. Yan, H. et al. Rapid Removal and Separation of Iron(II) and Manganese(II) from Micropolluted Water Using Magnetic Graphene Oxide. *ACS Appl. Mater. Interfaces* **6**, 9871–9880 (2014).

Changes made:

(1) We added a description and figure for the adsorption kinetic parameters of Al^{3+} during the ion desorption of Co^{2+} , Mn^{2+} , and Sr^{2+} by a pseudo-second-order rate model in the revised manuscript (pages 7–8 and Fig. 2c):

“We analysed the kinetic parameters of the ion desorption. Considering that the ion desorption occurred via Al^{3+} ion substitution, the desorption kinetic parameters of Co^{2+} , Mn^{2+} , and Sr^{2+} can be estimated by the adsorption kinetic parameters of Al^{3+} during the desorption processes. A pseudo-second-order rate equation³⁴, which has been widely applied to the adsorption of grapheme-based materials²⁷, was applied in the ion desorption via Al^{3+} ion substitution as follows:

$$\frac{t}{q_t} = \frac{1}{kq_e^2} + \frac{1}{q_e}t \quad (1)$$

where k ($g/mg \text{ min}^{-1}$) is the equilibrium rate constant of the pseudo-second-order rate for Al^{3+} , q_t (mg/g) is the amount of Al^{3+} adsorbed on the M-GO at time t (min), and q_e (mg/g) is the adsorption capacity at equilibrium.”

“As shown in Fig. 2c, the calculated adsorption capacities ($q_{e \text{ cal}}$) are consistent with the corresponding experimental values ($q_{e \text{ exp}}$), and the R^2 for the linear plots are close to 1, indicating that the kinetic adsorption can be well described by the pseudo-second-order rate equation. Remarkably, the k values of Al^{3+} during the desorption of the Co^{2+} , Mn^{2+} , and Sr^{2+} ions are 55.3, 51.3, and 41.8 $g/mg \text{ min}^{-1}$, respectively, which are about two or three orders of magnitude higher than the equilibrium rate constants of other types of adsorbents, including zeolites, zinc ferrite nanoparticles, biochar, GO-based membranes, and polymeric adsorbents (Supplementary Table 1). The results indicate the ultrafast Al^{3+} ion substitution adsorption, as well as the simultaneous ion desorption of Co^{2+} , Mn^{2+} , and Sr^{2+} on M-GO.”

(2) We added a table in the revised Supplementary Information (Section 4) that compares the equilibrium rate constant (k) of the pseudo-second-order rate on different adsorbents in solutions for multivalent metal ions in the literature (Supplementary Table 1):

“Section 4: Comparison of the equilibrium rate constant (k) of the pseudo-second-

order rate on different adsorbents in solutions for multivalent metal ions in the literature.

Supplementary Table 1. Comparison of the equilibrium rate constant (k) of the pseudo-second-order rate on different adsorbents in solutions for multivalent metal ions in the literature.”

Adsorbents	Target ions	k (g/mg min⁻¹)	References
ZIF-8	Cu ²⁺	2.81 × 10 ⁻¹	Ref. ⁵
ZF-NPS	Cd ²⁺	1.65 × 10 ⁻¹	Ref. ⁶
BC@MnO ₂	Cu ²⁺	4.60 × 10 ⁻³	Ref. ⁷
	Cu ²⁺	5.58 × 10 ⁻³	Ref. ⁸
MoS ₄ -LDH	Pb ²⁺	2.71 × 10 ⁻¹	Ref. ⁸
	Hg ²⁺	3.62 × 10 ⁻¹	Ref. ⁸
Chitosan	Al ³⁺	1.68 × 10 ⁻³	Ref. ⁹
GO	Cu ²⁺	1.55 × 10 ⁻²	Ref. ¹⁰
membranes	Cd ²⁺	1.79 × 10 ⁻²	Ref. ¹⁰
	Ni ²⁺	1.05 × 10 ⁻²	Ref. ¹⁰
GO/cellulose	Co ²⁺	9.40 × 10 ⁻³	Ref. ¹¹
membranes	Zn ²⁺	9.70 × 10 ⁻³	Ref. ¹¹
	Pb ²⁺	5.50 × 10 ⁻³	Ref. ¹¹
VA-PG	UO ₂ ²⁺	9.10 × 10 ⁻⁷	Ref. ¹²
GO-MnFe ₂ O ₄	Pb ²⁺	9.50 × 10 ⁻¹	Ref. ¹³
PANI/GO	Sr ²⁺	9.50 × 10 ⁻³	Ref. ¹⁴
MGO-PP	UO ₂ ²⁺	1.90 × 10 ⁻³	Ref. ¹⁵
M-RGO	As ³⁺	5.00 × 10 ⁻²	Ref. ¹⁶
	Co ²⁺	1.86 × 10 ⁻²	Ref. ²
	Cu ²⁺	1.10 × 10 ⁻³	Ref. ¹⁷
	Fe ²⁺	47.2	Ref. ³
M-GO	Mn ²⁺	43.6	Ref. ³
	Al ³⁺ (in Co ²⁺ solution)	55.3	this work
	Al ³⁺ (in Mn ²⁺ solution)	51.3	this work
	Al ³⁺ (in Sr ²⁺ solution)	41.8	this work

3. (1) The binding of Al³⁺, Co (II), Mn (II), and Sr (II) on M-GO should be calculated in detail, especial the bond distance, the electronic density etc. (2) The theoretical calculation about the kinetic sorption of Al³⁺ on M-GO should be added to understand the rapid sorption of Al³⁺ on M-GO.

Reply: We thank the referee for his/her constructive suggestions.

(1) According to the referee’s suggestions, we re-performed the DFT calculations on X@G(G=C₆₈H₂₂, X=Al³⁺-(H₂O)₆, Co²⁺-(H₂O)₆, Mn²⁺-(H₂O)₆, and Sr²⁺-(H₂O)₆). To

obtain more reliable results, a variety of possible initial structures were considered, and M06-2X/Def2-SVP was employed to calculate the optimized geometries, energies, and frequencies. As shown in Fig. R3, all the hydrated cations can be adsorbed on the surface of G, and the adsorption distances range from 2.38 Å to 2.73 Å. The corresponding adsorption energies of X@G listed in Fig. R4 show the strong hydrated cation– π interactions between the hydrated cations and the aromatic rings, which are much stronger than those of alkali metal ions in our early reports [*Nature* **550**, 380–383 (2017); *Nat. Chem.* **10**, 776–779 (2018); *Phys. Chem. Chem. Phys.* **21** (14), 7623–7629 (2019)]. The adsorption energies of $\text{Co}^{2+}\text{-(H}_2\text{O)}_6\text{@G}$, $\text{Mn}^{2+}\text{-(H}_2\text{O)}_6\text{@G}$, and $\text{Sr}^{2+}\text{-(H}_2\text{O)}_6\text{@G}$ are very close (around -80 kcal/mol), while the adsorption energy of $\text{Al}^{3+}\text{-(H}_2\text{O)}_6\text{@G}$ is approximately 75% higher (-139 kcal/mol). This is also supported by the results of cation partial charges (δ) and the electron density differences of X@G, where Al^{3+} leads to the greatest reduction in partial charges and the greatest increase in electron densities mainly transferred from $\text{C}_{68}\text{H}_{22}$ (see Fig. R3). Clearly, these results reveal the strong advantages of Al^{3+} ion adsorption on M-GO compared to Co^{2+} , Mn^{2+} , and Sr^{2+} ions.

Fig. R3. The most stable optimized geometries and electron density differences of (a) $\text{Al}^{3+}\text{-(H}_2\text{O)}_6\text{@G}$, (b) $\text{Co}^{2+}\text{-(H}_2\text{O)}_6\text{@G}$, (c) $\text{Mn}^{2+}\text{-(H}_2\text{O)}_6\text{@G}$, and (d) $\text{Sr}^{2+}\text{-(H}_2\text{O)}_6\text{@G}$ at the level of M06-2X/Def2-SVP. Spheres in green, white, red, pink, blue, purple, and yellow represent C, H, O, Al^{3+} , Co^{2+} , Mn^{2+} , and Sr^{2+} , respectively. Adsorption distances (in Å) and partial charges of cation δ (in atomic units) are listed. The increased and decreased electron densities (from -0.1 to 0.1) are in khaki and grey, respectively.

Fig. R4. Adsorption energies of X@G at the level of M06-2X/Def2-SVP.

(2) We further performed AIMD simulations on the kinetic process of Al^{3+} adsorption on the graphene surface. Figure R5a shows the initial system containing a graphene sheet, 83 water molecules, 3 Cl^- anions, and 1 Al^{3+} cation. Initially, the Al^{3+} cation is placed 0.80 nm from the graphene sheet. To rapidly minimize the approximate energy of the system, we performed a 200-step geometry optimization with the Al^{3+} cation fixed. Then, based on the energy minimization result, we performed six AIMD simulations. Both the geometry optimization and AIMD simulations were performed using the CP2K 9.1 package, through the PBE method and DZVP-GTH basis. During the AIMD simulations, we placed a position constraint on the 3 Cl^- anions, so that the Al^{3+} would not be attracted by the Cl^- anions and approach them. The temperature was set at 300 K by Nosé-Hoover thermostat.

Fig. R5. AIMD simulations on the kinetic process of Al³⁺ adsorption on the graphene surface. **(a)** Initial system. The spheres in red, white, and cyan represent the O, H, and C atoms, respectively, and pink and green represent the Al³⁺ and Cl⁻ ions. **(b)** Al³⁺ cation adsorbed on the graphene. The distance between the Al³⁺ cation and graphene sheet is < 0.35 nm. **(c)** Adsorption time for each AIMD simulation. **(d)** MSD for Al³⁺ in the direction perpendicular to the graphene and water molecules.

Based on the simulation results, we recorded the adsorption time for the Al³⁺ cation. The adsorption criterion used here is an adsorption distance less than 0.35 nm, which is the height of one water layer [*J. Chem. Phys.* **138**, 054117 (2013); *Nano Lett.* **15**, 5744 (2015)]. From Fig. R5c, the adsorption time in six trajectories ranges from 10.7 to 28.5 ps. We further calculated the MSD of the Al³⁺ cation and water molecules along the direction perpendicular to the graphene (Fig. R5d), where the slope of the Al³⁺ fitted curves is about 30 times that of the water molecules. The results show a fast adsorption speed of Al³⁺ to graphene. Thus, this fast adsorption speed, together with the huge specific surface area and the high dispersibility of M-GO, induce the rapid sorption of Al³⁺ on M-GO in solution.

Changes made:

We added the most stable structures, adsorption energies, cation adsorption distances, cation partial charges, and electron density differences of all systems, with the corresponding discussions and theoretical calculation method, to the revised manuscript (Fig. 5 and pages 10, 11, and 13) and Revised Supplementary Information Section 7 (Supplementary Fig. 6):

“We further performed quantum chemistry calculations to elucidate the underlying physical mechanism occurring on the surface of graphene. We used the hydrocarbon C₆₈H₂₂ as a model for graphene and calculated the corresponding hydrated cation adsorbed complexes X@G (G=C₆₈H₂₂, X=Al³⁺-(H₂O)₆, Co²⁺-(H₂O)₆, Mn²⁺-(H₂O)₆, and Sr²⁺-(H₂O)₆) at the level of M06-2X/Def2-SVP. Here, Al³⁺-(H₂O)₆@G and Co²⁺-(H₂O)₆@G are chosen as examples, and Fig. 5 shows their structures, cation adsorption distances, cation partial charges, electron density differences, and adsorption energies. For Mn²⁺-(H₂O)₆@G and Sr²⁺-(H₂O)₆@G, the corresponding results are shown in Supplementary Fig. 6. Calculation results show that all hydrated ions can be stably adsorbed on the surface of G, and the adsorption distances range from 2.38 Å to 2.73 Å. The adsorption energies of Co²⁺-(H₂O)₆@G, Mn²⁺-(H₂O)₆@G, and Sr²⁺-(H₂O)₆@G are very close (around -80 kcal/mol), while the adsorption energy of Al³⁺-(H₂O)₆@G is approximately 75% higher (-139 kcal/mol). This is also supported by the results for the cation partial charges (δ) and electron density differences of X@G, showing that Al³⁺ leads to the greatest reduction in partial charges and the greatest increase in electron densities mainly transferred from C₆₈H₂₂. Clearly, these results reveal the strong advantages of Al³⁺ ion adsorption on M-GO compared to Co²⁺, Mn²⁺, and Sr²⁺ ions.”

“Theoretical calculation. The M06-2X⁴³ method and Def2-SVP basis set⁴⁴ were employed for geometric optimization, frequency, and energy calculations. Both the low-spin state and high-spin state for Co²⁺ and Mn²⁺ systems were considered. All minima have no virtual frequency. The adsorption energies (ΔE_X) are defined as follows:

$$\Delta E = E_{X@G} - E_G - E_X, \quad (3)$$

where $E_{X@G}$ denotes the total energy of the cation/hydrated cation adsorbed on graphene, and E_G and E_X denote the energies of the isolated graphene and the cation/hydrated cation, respectively. Partial charges (δ) at the M06-2X/def2-SVP level were calculated using natural bond orbital (NBO) analysis^{45,46}. All electronic

calculations were performed using the Gaussian 16 program package⁴⁷. The electron density differences were analysed using the Multiwfn program⁴⁸, and the structures were visualized using VMD⁴⁹.”

4. It is very strange that the sorption of Sr^{2+} is the highest, but the desorption percentage is also the highest.

Reply: We thank the referee for his/her comments.

To respond to the referee’s comments, we checked and further repeated the experiments (Fig. R6), obtaining results that are consistent with the results in our main text (Fig. 3).

Fig. R6. Ion adsorption and desorption of M-GO (repeated experiments). (a) Equilibrium adsorption capacity of M-GO for Co^{2+} , Mn^{2+} , and Sr^{2+} . (b) Desorption rate of Co^{2+} , Mn^{2+} , and Sr^{2+} from M-GO by the subsequent addition of 6 mg/L Al^{3+} .

In the previous version, the adsorption capacity is evaluated in mg/g (Fig. R7a). It should be noted that, compared with the adsorption capacity in mg/g, the adsorption capacity in mmol/g takes into account the number of ions adsorbed by M-GO with a certain specific surface area, and this can provide a more accurate adsorption difference between ions. As can be seen from Fig. R7b, the adsorption performances of M-GO for the three cations are very similar. This also can be confirmed by our DFT computations, showing that the corresponding adsorption energies of X@G are very close (Fig. R4). For Sr^{2+} , the relatively low adsorption capacity in mmol/g and adsorption energy correspond to a higher desorption rate.

Fig. R7. Ion adsorption and desorption of M-GO in (a) mg/g and (b) mmol/g.

Referee #2 (Remarks to the Author):

This paper by Chen and co-workers reports efficient desorption of adsorbed metal ions on magnetite-graphene oxide (M-GO) by addition of Al³⁺ ions. Usefulness of the reported method has been demonstrated by concentration of radioactive ⁶⁰Co through simple filtration and addition of Al³⁺ ions processes. These findings are of interest, but the biggest concern from a viewpoint of practical use and sustainability is the reusability of M-GO materials after desorption with Al³⁺ ions. (1) As indicated by competition experiments and theoretical calculations, Al³⁺ ions strongly bind onto M-GO over other metal ions such as Co²⁺. Then, how can adsorbed Al³⁺ ions be desorbed from M-GO? If it requires addition of a large amount of HCl or NaOH, a desorption problem that the authors claimed in the introduction section will occur again. (2) In addition, desorption selectivity upon addition of Al³⁺ ions should be discussed. Namely, analyze desorption amount of metal ions using M-GO which adsorbed two (or more) different metal ions.

Reply: We sincerely thank the referee for these constructive suggestions and very positive comments regarding the novelty of our manuscript.

(1) The referee raised an important concern on the cycle sustainability of the M-GO after desorption with Al³⁺ ions. Clearly, it is a greater challenge to achieve Al³⁺ ion desorption, as the higher trivalent metal ions of Al³⁺ interact more strongly with the M-GO sheet than bivalent metal ions.

Fortunately, we found that the Al³⁺ ions adsorbed on M-GO can be effectively desorbed by the addition of a small amount of NH₃·H₂O. In detail, 75 μL NH₃·H₂O (25~28%) was added to the 30 mL mixture solutions to adjust the pH to 10, and then the mixture underwent magnetic separation and filtration. The separated M-GO was desorbed again with 30 mL DI water (containing 75 μL NH₃·H₂O). The concentrations of Al³⁺ in the filtrates were determined. The desorption rates of Al³⁺ on M-GO reached

78.5 ± 4.0% and 99.9 ± 0.1% for the two desorption steps, indicating that recycled M-GO could be achieved. However, with the same amount of NH₃·H₂O added to Co²⁺@M-GO, the desorption of Co²⁺ ions cannot be achieved. Importantly, the recycled M-GO could be reused easily multiple times without compromising its adsorption efficiency and magnetic performance (Fig. R8). Furthermore, the corresponding concentration of NH₃·H₂O used here was two to three orders of magnitude smaller than that for the conventional acid-base desorption method.

We noted that alkaline conditions generally increase the ion adsorption of GO [*J. Phys. Chem. C* **115**, 25234-25240 (2011); *ACS Appl. Mater. Interfaces* **6**, 9871-9880 (2014)]. However, Al³⁺ is unique under alkaline conditions, where aluminate anion [Al(OH)₄]⁻ is the dominant species at pH 10 [*Sci. Total Environ.* **662**, 1003–1011 (2019); *Int. J. Environ. Anal. Chem.* **86**, 1007–1018 (2006)]. Such anionic species would exhibit weak adsorption or repulsion to an electronegative π-conjugated system, including graphene, GO, and other materials composed of aromatic rings [*Chem. Rev.* **97**, 1303-1324 (1997); *Nat. Chem.* **10** (7), 776-779 (2018); *Sci. Rep.* **3**, 3436 (2013)].

Fig. R8. Al³⁺ ion desorption and reusability of M-GO for Co²⁺ adsorption. (a) Desorption of Al³⁺ on Al³⁺@M-GO solution. (b) Re-adsorption of recycled M-GO for 10 mg/L of Co²⁺. Error bars indicate the standard deviation from three different samples.

(2) Based on the referee's suggestions, we performed new experiments on the adsorption kinetics of mixed Co²⁺, Mn²⁺, and Sr²⁺ salt solutions by M-GO and the desorption kinetics of Al³⁺. As shown in Fig. R9, rapid ion adsorption of the mixed solution adsorbed by M-GO occurred within 1 min. The corresponding equilibrium adsorption capacities were 3.4 ± 0.1, 1.3 ± 0.1, and 2.0 ± 0.1 mg/g for Co²⁺, Mn²⁺, and Sr²⁺, respectively. With the subsequent addition of 10 mg/L Al³⁺ ions at 60 min, thorough desorption of Co²⁺, Mn²⁺, and Sr²⁺ ions originally adsorbed on M-GO

occurred within ~ 1 min, along with the corresponding rapid adsorption of Al^{3+} ions. The total desorption rate for all mixed ions was $98.6 \pm 1.3\%$, which corresponds to desorption rates of $98.6 \pm 1.6\%$, $99.9 \pm 0.1\%$, and $97.3 \pm 4.7\%$ for Co^{2+} , Mn^{2+} , and Sr^{2+} , respectively. Therefore, the rapid adsorption, especially the efficient mixed ions desorption on M-GO by adding very low amounts of Al^{3+} , is still consistent with those of the single-salt solutions.

We further performed DFT calculations on $\text{X}@G$ ($G=\text{C}_{68}\text{H}_{22}$, $\text{X}=\text{Al}^{3+}-(\text{H}_2\text{O})_6$, $\text{Co}^{2+}-(\text{H}_2\text{O})_6$, $\text{Mn}^{2+}-(\text{H}_2\text{O})_6$, and $\text{Sr}^{2+}-(\text{H}_2\text{O})_6$) to obtain optimized geometries, energies, and frequencies. As shown in Fig. R9, all hydrated cations can be adsorbed on the surface of the aromatic rings. The adsorption energies of $\text{Co}^{2+}-(\text{H}_2\text{O})_6@G$, $\text{Mn}^{2+}-(\text{H}_2\text{O})_6@G$, and $\text{Sr}^{2+}-(\text{H}_2\text{O})_6@G$ are very close (around -80 kcal/mol), revealing the strong and comparable ion adsorption of Co^{2+} , Mn^{2+} , and Sr^{2+} on the aromatic rings. Meanwhile, the adsorption energy of $\text{Al}^{3+}-(\text{H}_2\text{O})_6@G$ is approximately 75% higher (-139 kcal/mol), revealing the strong advantages of Al^{3+} ion adsorption on M-GO compared to Co^{2+} , Mn^{2+} , and Sr^{2+} ions.

Fig. R9. (a) Adsorption kinetics of the mixed solution (10 mg/L Co^{2+} , 10 mg/L Mn^{2+} , and 10 mg/L Sr^{2+}) by M-GO, as well as adsorption kinetics of 10 mg/L Al^{3+} added to the mixed solutions at 298 K. (b) Unit conversion in (a) from mg/g to mmol/g. Error bars indicate the standard deviation from three different samples.

Changes made:

(1) We added a description and figure for the Al^{3+} ion desorption and the reusability of M-GO for Co^{2+} adsorption in the revised manuscript (pages 2, 3, 10, and 13 and Fig. 4) and the corresponding detailed operations in the revised Supplementary Information (Section 1):

“Very importantly, the M-GO could be recycled and reused easily without compromising its adsorption efficiency and magnetic performance, based on the unique hydration anionic species of aluminium ions under alkaline conditions.”

“The Al^{3+} ions adsorbed on M-GO can be effectively desorbed through the addition of a small amount of $NH_3 \cdot H_2O$. We added 75 μL $NH_3 \cdot H_2O$ (25~28%) to the ~30 mL of $Al^{3+}@M-GO$ mixture solution (pH ~10) and were able to recycle and reuse M-GO without compromising its adsorption efficiency and magnetic performance.”

“From the cycle sustainability of $Al^{3+}@M-GO$, the higher trivalent metal ions of Al^{3+} interact strongly with the M-GO sheet over other bivalent metal ions, which introduced a more difficult desorption. Fortunately, we found that the Al^{3+} ions adsorbed on M-GO can be effectively desorbed adding a small amount of $NH_3 \cdot H_2O$. In detail, 75 μL $NH_3 \cdot H_2O$ (25~28%) was added to the 30 mL mixture solutions to adjust the pH to 10, and then the mixtures were separated through magnetic separation and filtration. The separated M-GO was desorbed again with 30 mL DI water (containing 75 μL $NH_3 \cdot H_2O$). The concentrations of Al^{3+} in the filtrates were determined. The desorption rates of Al^{3+} on M-GO reached $78.5 \pm 4.0\%$ and $99.9 \pm 0.1\%$ for the two desorption steps, indicating achievement of recycled M-GO. However, when adding the same amount of $NH_3 \cdot H_2O$ to $Co^{2+}@M-GO$, the desorption of Co^{2+} ions cannot be achieved. Importantly, the recycled M-GO can be reused easily multiple times without compromising its adsorption efficiency and magnetic performance (Fig. 4). Furthermore, the corresponding concentration of $NH_3 \cdot H_2O$ used here was two to three orders of magnitude smaller than the conventional acid-base desorption method.”

“We noted that the alkaline conditions generally increase the ion adsorption of $GO^{31,32}$. However, Al^{3+} is unique under alkaline conditions, where aluminate anion $[Al(OH)_4]^-$ will be the dominant species at pH 10^{39,40}. Such anionic species would exhibit weak adsorption or repulsion to an electronegative π -conjugated system, including graphene, GO, and other materials composed of aromatic rings^{15,22,26}.”

“Notably, based on the unique hydrolysis of Al^{3+} , the M-GO can be conveniently recycled and easily reused multiple times without compromising its adsorption efficiency and magnetic performance.”

(2) We added a description of desorption selectivity upon addition of Al^{3+} ions in

the revised manuscript (page 8) and a corresponding figure in the revised Supplementary Information Section 5 (Supplementary Fig. 4):

“In addition, we analysed the adsorption kinetics of mixed Co^{2+} , Mn^{2+} , and Sr^{2+} salt solutions by M-GO and the desorption kinetics of Al^{3+} . A similar rapid ion adsorption of the mixed solution adsorbed by M-GO occurred within 1 min (Supplementary Fig. 4). The corresponding equilibrium adsorption capacities were 3.4 ± 0.1 , 1.3 ± 0.1 , and 2.0 ± 0.1 mg/g for Co^{2+} , Mn^{2+} , and Sr^{2+} , respectively. With the subsequent addition of 10 mg/L Al^{3+} ions at 60 min, the thorough desorption of Co^{2+} , Mn^{2+} , and Sr^{2+} ions originally adsorbed on M-GO occurred within ~1 min, along with the corresponding rapid adsorption of Al^{3+} ions. The total desorption rate for all mixed ions was $98.6 \pm 1.3\%$, which corresponds to desorption rates of $98.6 \pm 1.6\%$, $99.9 \pm 0.1\%$, and $97.3 \pm 4.7\%$ for Co^{2+} , Mn^{2+} , and Sr^{2+} , respectively. Therefore, the rapid adsorption, especially the efficient mixed ions desorption on M-GO by adding very low amounts of Al^{3+} , is still consistent with those of the single-salt solutions.”

Other points:

Phrases such as “adding trace amounts of Al^{3+} ” (page 3, line 4), “a negligible volume of Al^{3+} ” (page 5, line 7) are overstatements. Analytically, these amounts are not negligible.

Reply: We thank the referee for his/her comments.

In response to the referee’s comments, we revised “trace amounts” by “very low amounts” and added a detailed description for “very low amounts” and “a negligible volume” in our revision. In our experiments, the volume ratio of the Al^{3+} solution to the mixed solution was 1:500. For example, 60 μL of Al^{3+} was added to 30 mL mixed solution (10 mg/L Al^{3+} in the prepared mixed salt solution) or 400 μL of Al^{3+} was added to 200 mL mixed solution (10 mg/L Al^{3+} in the prepared mixed salt solution).

Changes made:

We revised “trace amounts” by “very low amounts” and added a detailed description of “very low amounts” and “a negligible volume” in the revised manuscript (page 3, 5, and 6) and revised Supplementary Information (Section 1):

“In this study, we observed the unexpectedly rapid and efficient desorption of ions adsorbed on M-GO by adding very low amounts of Al^{3+} (at a volume ratio of 1:500).”

“In Step 3, 60 μ L of Al^{3+} solution was subsequently added such that the concentration of Al^{3+} in the mixtures was 20 mg/L. The solution was stirred at 298 K for 5 min, and then separated through magnetic separation and filtration.”

“The prepared M-GO (200 mg) was added to 200 mL solutions of 10 mg/L Co^{2+} , Mn^{2+} , and Sr^{2+} , and the solutions were stirred at 298 K for 125 min. Then, a negligible volume of highly concentrated Al^{3+} solution (400 μ L) was subsequently added such that the concentration of Al^{3+} in the mixtures was 10 mg/L. The mixtures were then stirred at 298 K for another 125 min. At designated time intervals ranging from 0 to 250 min, 5 mL of the solution was taken at each interval for filtration separation and measurement of the residual ion concentration.”

(2) References cited in the introduction section are a bit biased. The authors cite only uranium examples for general high-efficiency extraction and concentration. Furthermore, references 6–10 are papers discussing cation– π interactions or noncovalent interactions, but not showing graphene-based materials.

Reply: We thank the referee for his/her comments.

To respond to the referee’s suggestion, we have cited the relevant papers in the introduction of our revised manuscript.

Changes made:

We cited relevant papers in the introduction of our revised manuscript (Page 2).

“Adsorption separation technology is one of the most effective and economical separation methods for high-efficiency extraction^{1–4}, concentration^{5–7}, and purification^{8–11}. Ion-surface adsorption between cations and graphene-based materials^{12–18} results in strong adsorption due to the one-atom-layer thickness and perfect aromatic ring structure of graphene¹⁹.”

Referee #3 (Remarks to the Author):

I recommend rejecting this paper based on poor English grammar usage. The manuscript in its current form is substandard. I am not commenting on the scientific merits of the research; rather, I am unable to understand the paper sufficiently to judge its scientific merits. The authors are encouraged to submit this manuscript for

professional English technical editing by a fluent speaker.

Reply: We thank the referee for his/her comments.

Our revised manuscript has been edited by a professional English editing service (San Francisco Edit).

We sincerely thank the referees for their very helpful suggestions and very constructive comments, which helped us to improve our manuscript. In accordance with the suggestions of the referees, we added new calculations involving the binding of the cations on M-GO and the kinetic sorption of Al^{3+} . We found that the interaction between Al^{3+} and graphene is the strongest, and Al^{3+} can be adsorbed very rapidly as seen from the dynamic simulations. Notably, we experimentally demonstrated the rapid and efficient desorption of typical bivalent ions on M-GO by adding very low amounts of Al^{3+} and further were able to conveniently reuse M-GO based on the unique hydrolysis of Al^{3+} . Our new experimental results confirmed the efficient ion desorption of graphene-based materials with a method that is facile, convenient, and reusable and consumes low amounts of reagents. This advancement will greatly improve the adsorption applications of Al^{3+} ion desorption in energy, biology, environment, and materials science fields.

We think that all ambiguities in the manuscript have been clarified. We hope that the referees will be satisfied with our response and will now recommend its publication in *Nature Communications*.

REVIEWER COMMENTS

Reviewer #1 (Remarks to the Author):

The authors revised the manuscript carefully. I recommend for publication.

Reviewer #2 (Remarks to the Author):

The authors have addressed most of the points that reviewers pointed out in the previous round. Desorption of Al^{3+} ions using ammonia solution demonstrates a good reusability and sustainability. However, this manuscript still leaves several points to be addressed. Another concern arose in this revised manuscript is chemical stability of magnetite–graphene oxide. Do the Fe_3O_4 nanoparticles remain on M-GO during recycling process? Please show the magnetic properties of M-GO such as Fig. 1d after the adsorption–desorption cycles.

Supplementary Table 1 which was added in this revision provides a good comparison. But, to show the scope of M-GO as a reusable adsorbent, it is also important to investigate equilibrium rate constants in Cu^{2+} and Cd^{2+} solutions along with their adsorption/desorption rates.

In addition, make sure that figure numbers quoted in the manuscript are correct: for example, in lines 90 and 92 on page 4 'Fig. 1c' should be Fig. 1d.

Responses to Referees' Comments

Manuscript ID: **NCOMMS-21-21067A-Z**

Title: **"Unexpectedly efficient ion desorption of graphene-based materials"**

Referees' Comments

Referee #1 (Remarks to the Author):

The authors revised the manuscript carefully. I recommend for publication.

Reply: Many thanks.

Referee #2 (Remarks to the Author):

The authors have addressed most of the points that reviewers pointed out in the previous round. Desorption of Al³⁺ ions using ammonia solution demonstrates a good reusability and sustainability. However, this manuscript still leaves several points to be addressed.

Reply: We sincerely thank the referee for his/her very positive comments.

1. Another concern arose in this revised manuscript is chemical stability of magnetite-graphene oxide. Do the Fe₃O₄ nanoparticles remain on M-GO during recycling process? Please show the magnetic properties of M-GO such as Fig. 1d after the adsorption-desorption cycles.

Reply: Thanks for the referee's comment and suggestion.

As suggested, we have added the high-resolution TEM images and magnetic properties of M-GO after each cycle. As shown in Fig. R1, the Fe₃O₄ nanocrystals were clear and well retained on each recycled M-GO. The corresponding saturation magnetization from their magnetization curves is 44.6, 50.1, and 46.2 emu/g, respectively, consistent with the original M-GO (Fig. 1d). These results demonstrate an excellent chemical stability of M-GO during recycling process.

Such good stability in magnetic performance allows M-GO to perform multiple magnetic separations in our recycling experiments.

Fig. R1. High-resolution TEM images and magnetic properties of M-GO after each cycle. The high-resolution TEM images (top) and magnetization curve at room temperature (down) for the M-GO after (a) 1 cycle, (b) 2 cycles, and (c) 3 cycles.

Changes made:

(1) We added the high-resolution TEM images and magnetization curves of M-GO after each cycle in the revised Supplementary Information Section 7 (Supplementary Fig. 6).

(2) We added a description in the revised manuscript (page 10):

“Importantly, the recycled M-GO can be reused easily multiple times without compromising its adsorption efficiency and magnetic performance (Fig. 4 and Supplementary Fig. 6).”

2. Supplementary Table 1 which was added in this revision provides a good comparison. But, to show the scope of M-GO as a reusable adsorbent, it is also important to investigate equilibrium rate constants in Cu^{2+} and Cd^{2+} solutions along with their adsorption/desorption rates.

Reply: Thanks for the referee’s suggestions.

According to the referee’s suggestions, we further performed ion adsorption and desorption kinetics experiments of M-GO for Cu^{2+} and Cd^{2+} . As shown in Fig. R2a, rapid ion adsorption of Cu^{2+} and Cd^{2+} adsorbed by M-GO occurred within 1 min after adding ions, and the equilibrium adsorption capacities of M-GO are 6.0 ± 0.2 and 5.9

± 0.2 mg/g for Cu^{2+} and Cd^{2+} , respectively. Then a rapid desorption of Cu^{2+} and Cd^{2+} ions was observed when 10 mg/L Al^{3+} ions were added at 125 min. The corresponding desorption rate were $61.1 \pm 2.5\%$ and $97.3 \pm 2.3\%$ for Cu^{2+} and Cd^{2+} , respectively (Fig. R2b). We note that the adsorption energy of $\text{Cu}^{2+}@G$ is much higher than those of other divalent ions (Co^{2+} , Mn^{2+} , Sr^{2+} , and Cd^{2+}) [*Sci. Rep.* **3**, 3436 (2013); *Appl. Surf. Sci.* **546**, 149110 (2021)], resulting in the moderate desorption of Cu^{2+} under the same conditions. However, a satisfied desorption of Cu^{2+} can be obtained by successive addition of Al^{3+} ions, for example, the desorption rate of Cu^{2+} can reach $82.9 \pm 0.2\%$ after the second desorption cycle, yielding the desorption performance still superior to those of conventional desorption methods.

Fig. R2. Ion adsorption and desorption of M-GO for Cu^{2+} and Cd^{2+} . (a) Adsorption kinetics of 10 mg/L Cu^{2+} and Cd^{2+} , as well as adsorption kinetics of 10 mg/L Al^{3+} added to the salt solutions (Cu^{2+} and Cd^{2+}) at 298 K, respectively. q_t denotes the adsorption capacity of M-GO with time. (b) Desorption rate of Cu^{2+} and Cd^{2+} from M-GO by the subsequent addition of Al^{3+} . (c) Adsorption kinetic parameters of Al^{3+} during the ion desorption of Cu^{2+} and Cd^{2+} by a pseudo-second-order rate model. Error bars indicate the standard deviation from three different samples.

The calculated adsorption capacities ($q_{e \text{ cal}}$) listed in Fig. R2c, are consistent with the corresponding experimental values ($q_{e \text{ exp}}$), indicating that the kinetic adsorption can

be well described by the pseudo-second-order rate equation. The k values of Al^{3+} during the desorption of the Cu^{2+} and Cd^{2+} ions are 130.2 and 60.4 g/mg min^{-1} , respectively, which are much higher than the equilibrium rate constants of other types of adsorbents (Supplementary Table 1). We note that the k of Al^{3+} in the desorption process of Cu^{2+} is higher than those in other divalent ions (Co^{2+} , Mn^{2+} , Sr^{2+} , and Cd^{2+}), which we attribute to the less amount of Cu^{2+} desorption, allowing a shorter time to reach desorption equilibrium, thus resulting in a faster adsorption kinetic parameter. In all, these results indicate that the rapid desorption of the Cu^{2+} and Cd^{2+} ions also can be achieved by our method, indicating a wide application range of the method in this work.

Changes made:

(1) We added a description for the ion adsorption and desorption of M-GO for Cu^{2+} and Cd^{2+} in the revised manuscript (page 7):

“Furthermore, similar rapid desorption of the Cu^{2+} and Cd^{2+} ions also can be achieved by our method (see Supplementary Information, Section 4), showing a wide range of applications of the method in this work.”

(2) We added a corresponding detailed description and figure in the revised Supplementary (Section 4).

“The ion adsorption and desorption kinetics experiments of M-GO for Cu^{2+} and Cd^{2+} were further carried out, and the experimental procedures were consistent with those of Co^{2+} , Mn^{2+} , and Sr^{2+} , refer to Section 1. 10 mg/L Cu^{2+} and Mn^{2+} solutions were prepared with $\text{CuCl}_2 \cdot 2\text{H}_2\text{O}$ and $\text{CdCl}_2 \cdot 2.5\text{H}_2\text{O}$, respectively.

As shown in Supplementary Fig. 4a, rapid ion adsorption of Cu^{2+} and Cd^{2+} adsorbed by M-GO occurred within 1 min after adding ions, and the equilibrium adsorption capacities of M-GO are 6.0 ± 0.2 and 5.9 ± 0.2 mg/g for Cu^{2+} and Cd^{2+} , respectively. Then a rapid desorption of Cu^{2+} and Cd^{2+} ions was observed when 10 mg/L Al^{3+} ions were added at 125 min. The corresponding desorption rate were $61.1 \pm 2.5\%$ and $97.3 \pm 2.3\%$ for Cu^{2+} and Cd^{2+} , respectively (Supplementary Fig. 4b). We note that the adsorption energy of Cu^{2+} @G is much higher than those of other divalent ions (Co^{2+} , Mn^{2+} , Sr^{2+} , and Cd^{2+})^{6,7}, resulting in the moderate desorption of Cu^{2+} under the same conditions. However, a satisfied desorption of Cu^{2+} can be obtained by successive addition of Al^{3+} ions, for example, the desorption rate of Cu^{2+} can reach $82.9 \pm 0.2\%$ after the second desorption cycle, yielding the desorption performance still superior to that of conventional desorption methods.

The calculated adsorption capacities ($q_{e\text{ cal}}$) listed in Supplementary Fig. 4c, are consistent with the corresponding experimental values ($q_{e\text{ exp}}$), indicating that the kinetic adsorption can be well described by the pseudo-second-order rate equation. The k values of Al^{3+} during the desorption of the Cu^{2+} and Cd^{2+} ions are 130.2 and 60.4 $g/mg\text{ min}^{-1}$, respectively, which are much higher than the equilibrium rate constants of other types of adsorbents (Supplementary Table 1). We noted that the k of Al^{3+} in the desorption process of Cu^{2+} is higher than that in other divalent ions (Co^{2+} , Mn^{2+} , Sr^{2+} , and Cd^{2+}), which we attribute to the less amount of Cu^{2+} desorption, allowing a shorter time to reach desorption equilibrium, thus resulting in a faster adsorption kinetic parameter. In all, these results indicate that the rapid desorption of the Cu^{2+} and Cd^{2+} ions also can be achieved by our method, indicating a wide application range of the method in this work.”

(3) The k values of Al^{3+} during the desorption of Cu^{2+} and Cd^{2+} ions were listed in the Supplementary Table 1.

3. In addition, make sure that figure numbers quoted in the manuscript are correct: for example, in lines 90 and 92 on page 4 ‘Fig. 1c’ should be Fig. 1d.

Reply: Many thanks. The errors have been corrected in the revised manuscript and we checked the manuscript clearly again.

Now, we think that all ambiguities in the manuscript have been clarified. We hope that the referee will be satisfied with our response and will now recommend its publication in *Nature Communications*.

REVIEWERS' COMMENTS

Reviewer #2 (Remarks to the Author):

The authors have answered all the points carefully with additional data. Now the discussions in the manuscript are quite convincing. I recommend publication in Nature Communications.